# Early Carboniferous Ostracods (Crustacea) from Death Valley, California, USA

**Mark A. S. McMenamin** 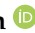

Department of Geology and Geography, Mount Holyoke College, South Hadley, MA 01075, USA; mmcmenam@mtholyoke.edu

**Abstract:** Silicified ostracods from the Tin Mountain Limestone provide new information regarding the Carboniferous paleontology of the Death Valley Region, California, USA. Acid maceration of marine limestones yielded the following ostracods: *Acratia* spp., *Bairdia quasilecta*, *Bairdia* sp. cf. *B. orientalis*, *Ceratobairdia* sp., *Kirkbya panamintensis* sp. nov., *Rectobairdia* sp. cf. *R. legumen*, and *Silenites* sp. This is the first report of *Ceratobairdia* and *Silenites* from the Tin Mountain Limestone. These ostracods occupied a carbonate ramp environment that formed during a major Paleozoic transgression. The ostracods played an important paleoecological role, likely as benthic marine scavengers in a shallow marine biotope along the northern shores of Pangea.

**Keywords:** ostracods; carboniferous; Mississippian; Tin Mountain Limestone; Death Valley; California; carbonate ramp; *Acratia*; *Bairdia*; *Ceratobairdia*; *Kirkbya*; *Rectobairdia*; *Silenites*





## 1. Introduction

Ostracods are a biostratigraphically important group of bivalved arthropods that have undergone a series of evolutionary radiations [1]. They remain important members of the aquatic modern arthropod fauna. Ostracod fossils have great utility for assessing paleobiogeographic isolation, particularly with regard to the breakup of Pangea. Ostracods can also help track global environmental changes through time [2]. In spite of their typically small size, silicified specimens have been recovered from Paleozoic and later limestones showing high-fidelity preservation. The objectives of this paper are to provide new information regarding the Carboniferous ostracods of the Death Valley region (including the first report of several Paleozoic ostracod genera from the region, plus the description of a new species), and to consider the depositional setting and paleoecological characteristics of this ancient ostracod fauna and its biotope.

## 2. Death Valley Ostracod Fossils

The Death Valley National Monument in California is a geological mecca, considered by many [3–6] to be a natural laboratory for the geosciences. New information regarding the paleontology and stratigraphy of the Death Valley region is of interest to professional geologists, educators, and geology students. The Tin Mountain Limestone figures prominently in the stratigraphic column of the Death Valley region [3,4,6].

An abundant fauna of ostracods occurs in the Lower Carboniferous Tin Mountain Limestone of eastern California (Table 1). Silicified ostracods were first reported from the Lower Carboniferous Tin Mountain Limestone of eastern California by Langenheim and Tischler [7] in a study of the Devonian and Mississippian paleontology of the Quartz Spring area, Inyo County. This study [7] was based on the pioneering geological mapping work of McAllister [8,9]. Sohn [10] described the distinctive ostracod *Shivaella macallisteri* from a suite of Tin Mountain Limestone samples collected by J. F. McAllister in 1965–1966. Sohn [10] and McAllister [9] used ostracod biostratigraphy to assign the Tin Mountain Limestone to the Kinderhookian or lowermost Osagean Series.

**Table 1.** Ostracod fossils illustrated in the literature from the Tin Mountain Limestone, eastern California, USA.

| Taxon | Reference |
|---|---|
| *Acratia* sp. A | [11]; this report |
| *Acratia* sp. B | this report |
| *Acratia* sp. C | this report |
| *Acratia* sp. D | this report |
| *Amphissites* sp. | [11] |
| *Bairdia quasilecta* | this report |
| *Bairdia* sp. cf. *B. orientalis* | this report |
| *Bairdia* sp. | [11] |
| *Ceratobairdia* sp. | this report |
| *Kirkbya panamintensis* sp. nov. | this report |
| *Rectobairdia* sp. cf. *R. legumen* | this report |
| *Shivaella macallisteri* | [10] |
| *Silenites* sp. | this report |

From the Tin Mountain Limestone collection 12858-PC (Funeral Mountains, I.G. Sohn identifications), McAllister [9] dated the lower Tin Mountain Limestone to the Tournaisian Stage (Kinderhookian-Osagean) based on the following ostracod taxa: Kinderhookian (*Psilokirkbyella ozarkensis?*; *Monoceratina?* n. sp. aff. *M.? elongata*); Kinderhookian-Osagean (*Tetrasacculus* sp. aff. *T. stewartae*; *Roundyella* n. sp. aff. *Scrobicula crestiformis*; *Kummerowia?* n. sp. aff. *Kirkbya fernglennensis*; *Kirkbyella* (*Berdanella*) n. sp. aff. *Kirkbyella annensis*; *Graphiadactylloides* n. sp. aff. *Graphiadactyllis moridgei*); and Osagean (*Amphissites* n. sp. aff. *A. similaris*; *Kummerowia?* n. sp. aff. *Kirkbya keiferi*; *Kirkbyella* (*Berdanella*) n. sp. aff. *Kirkbyella reticulata*; *Rectobairdia* sp. cf. *R. confragosa*, *Acratia* (*Cooperuna*) n. sp. aff. *Acratia* (*Cooperuna*) *similaris*; *Bohlenatia?* n. sp. aff. *Acanthoscaphia? banffensis*; and *Monoceratina* n. sp. aff. *M. virgata*). Unfortunately, no illustrations were provided for these taxa [9].

McMenamin and Witterschein [11] reported a suite of silicified ostracods in the Tin Mountain Limestone, collected by M. McMenamin under permit from the Death Valley National Monument at Lost Burro Gap (near lat 36°45′ long 117°30′). McMenamin and Witterschein [11] reported and illustrated the ostracods *Acratia* sp., *Amphissites* sp., and *Bairdia* sp., as well as the conodonts *Prioniodus* sp. *Polygnathus* cf. *P. symmetricus*, and *Siphonodella isosticha*.

The present study reports and describes *Acratia* spp., *Bairdia quasilecta*, *Bairdia* sp. cf. *B. orientalis*, *Ceratobairdia* sp., *Kirkbya panamintensis* sp. nov., *Rectobairdia* sp. cf. *R. legumen*, and *Silenites* sp. from the lower Tin Mountain Limestone. This is the first report of *Silenites* and *Ceratobairdia* from the Tin Mountain Limestone. The Tin Mountain ostracod fauna is the result of a dual evolutionary radiation of palaeocopid ostracods (*Amphissites*, *Kirkbya*) and podocopid ostracods (*Bairdia*, *Rectobairdia*, *Ceratobairdia*, *Silenites*, *Acratia*). Striking faunal similarity occurs between the Tin Mountain Limestone ostracod fauna and ostracods of comparable age from just south of the Kolyma River in the Russian Far East (stratigraphic sections at Kamenka and Dozhdlivyi Creeks [12]). As with the Tin Mountain fauna, the Kolyma ostracods show a dominance of palaeocopid ostracods (*Kirkbya*, *Amphissites*) and podocopid ostracods (*Acratia*, *Bairdia*) [12]. This suggests that these ostracod assemblages are, broadly speaking, part of the same fauna, a fauna that extended in shallow marine carbonate ramp habitats along the northern shores of Pangea at relatively low latitudes.

This ostracod fauna evidently extended from western North America to northwestern Siberia. As such, it represented part of a widespread Panthalassic carbonate ramp biotope, with the ostracods likely playing an important paleoecological role as benthic scavengers.

The distinctive shapes of the ostracods described here may reflect specialization for feeding on the remains of particular types of larger animals, such as echinoderms (crinoid remains abundant in the Tin Mountain Limestone), chordates, or large arthropods. Ostracods are known to feed on large arthropod carcasses, consuming the flesh from within the cuticle/carapace of the deceased animal. While scavenging in this way, the ostracods may congregate in large numbers. The Bowland Shale Formation (Upper Carboniferous) of Derbyshire, Great Britain, preserves thousands of specimens of nectobenthic ostracods (*Eocypridina carsingtonensis*) feeding on a shark carcass (*Orodus* sp.) [13].

## 3. Material and Methods

After being collected in stratigraphic sequence (Figure 1), limestone samples of Tin Mountain Limestone were broken into chunks and immersed in dilute hydrochloric and acetic acid. This process produced macerates containing silicified echinoderm fragments, bryozoans, brachiopods (some with well-preserved hinge areas), ostracods, and conodonts, the latter preserved as dark grey to black hydroxyapatite. Many of the ostracods (hundreds were encountered) were fragmentary, and many were tiny instars that were diagnostic to neither genus nor species. Several of the larger specimens, described here, were preserved as larger complete isolated valves. Several complete (i.e., both valves united) specimens were recovered. The ostracods were collected by sieve screening of the macerate using a 420-micron sieve, drying, and picking of the sample without use of heavy liquids.

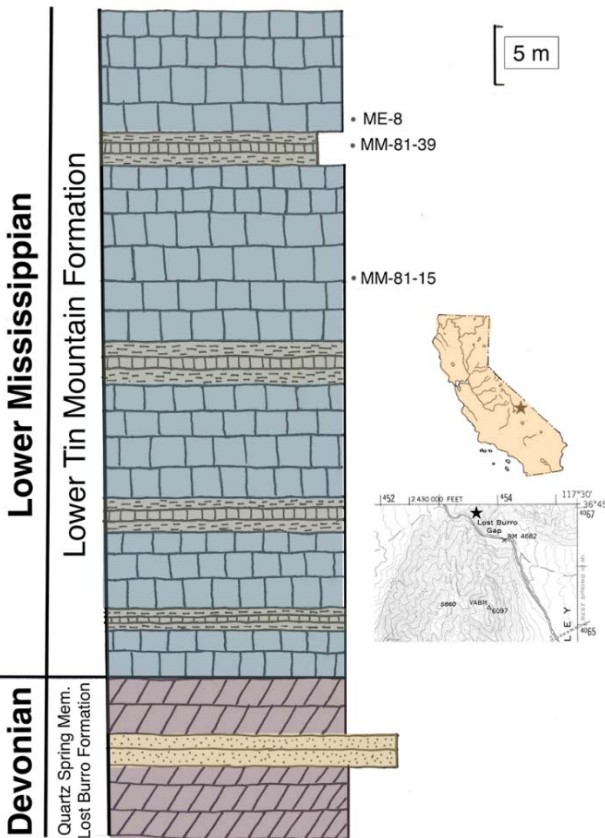

**Figure 1.** Stratigraphic section, north side of Lost Burro Gap, Inyo County, California. Upper inset map shows the location of the stratigraphic section on a map of California (star). Lower inset map shows the location of a section at Lost Burro Gap (star). Three ostracod-bearing fossil localities are indicated on the stratigraphic section (MM-81-15, MM-81-39 and ME-8). Key to the color scheme: *blue-grey*, limestone; *grey*, calcareous shales and thin-bedded limestones; *yellow*, dolomitic sandstone; *light purple*, dolostone. Scale bar for section = 5 m.

In addition to specimens recovered in the field, a raw sample (unpicked acid maceration residue) of Lower Tin Mountain Formation ostracods was obtained from the estate of Emeritus Professor Robert Lundin of Arizona State University, School of Earth and Space Exploration. The exact stratigraphic position of Lundin's sample is unknown, but the ostracod fauna it contains is comparable to that yielded by field sample MM-81-15. The latter was taken from approximately 29 m above the base of the Tin Mountain Limestone (Figure 1).

## 4. Stratigraphic Setting

The Tin Mountain Limestone (Lower Carboniferous/Lower Mississippian/Kinderhookian-early Osagean; 350–358.9 Ma) crops out in the southern Last Chance, northern Panamint, Argus, and Inyo Mountains of eastern California. The Tin Mountain Limestone is approximately 305 m thick, consisting largely of bluish-gray, skeletal mudstone/wackestone to micritic cherty pelmatozoan limestone (Figure 1). The formation represents a shallow marine limestone that was deposited in mudflats, lagoons, and offshore carbonate sand bars. It is rich in crinoids, corals (especially *Syringopora*, rugosans, and favositids), brachiopods (including *Brachythyris*, *Composita*, *Productella*, *Punctospirifer*, *Shumardella*, *Spirifer*), bryozoans, mollusks, ostracods, phillipsid trilobites, conodonts, foraminifera, and petrified wood. Calcareous fossils may develop partial to complete silicification.

The Tin Mountain Limestone was deposited as homoclinal ramp sediments developed during local transgressive conditions thought to have developed as part of tectonic downwarping due to crustal flexure during the creation of a nascent foreland basin as the Antler allochthon was emplaced [14]. These processes were linked to continent-wide subsidence and mantle downwelling associated with the assembly of Pangea [15,16]. The Tin Mountain Limestone was deposited during a time of an unusually high local sea level that has been attributed in part to continental margin subsidence resulting from supercontinent assembly [15,16].

The upper part of the Tin Mountain is basinal and developed spiculitic wackestones [14]. The transgressive aspect of the sequence at Lost Burro Gap can be seen in Figure 1, where sandy facies and dolostones of the Quartz Spring Member of the Lost Burro Formation (Upper Devonian, Famennian) are overlain by the lower Tin Mountain Formation. The Tin Mountain shows an interbedding of harmonically spaced calcareous shaly interbeds among cleaner limestone layers, providing possible evidence for cyclic sedimentation [17] that may have been linked to cyclothem development (Figure 1).

## 5. Systematic Paleontology

Morphological terms used here are based on Scott [18]. Abbreviations used here are as follows: AB = anterior border; ADB = antero-dorsal border; AVD = antero-ventral border; C = carapace; DB = dorsal border; H = height; L = length; LV = left valve; PB = posterior border; PDB = postero-dorsal border; PVB = postero-ventral border; RV = right valve; V = valve; VB = ventral border; W = width. Specimens with IGM numbers reside in the Institute of Geology Museum, Departmento de Paleontología, Cuidad Universitaria, México.

| |
|---|
| Phylum Euarthropoda Lankester, 1904 [19] |
| Class Ostracoda Latreille, 1802 [20] |
| Order Palaeocopida Henningsmoen, 1953 [21] |
| Suborder Beyrichicopina Scott, 1961 [22] |
| Superfamily Kirkbyacea Ulrich and Bassler, 1906 [23] |
| Family Kirkbyidae Ulrich and Bassler, 1906 [23] |
| Genus *Kirkbya* Jones, 1859 [24]<br>***Kirkbya panamintensis* nov. sp.**: Figures 2–5. |

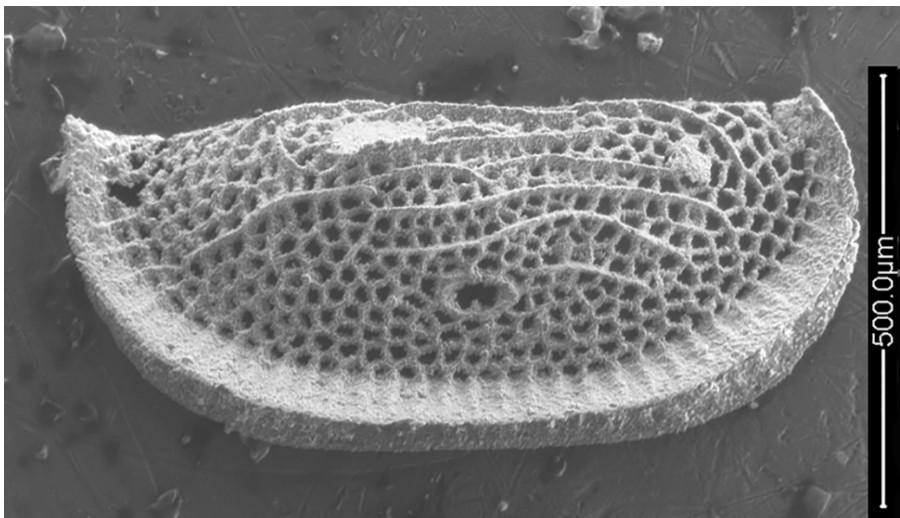

**Figure 2.** *Kirkbya panamintensis* nov. sp., right valve, lateral view. Holotype, IGM 5011(1). Scanning electron micrograph (secondary electrons). Lower Carboniferous (Lower Mississippian), Lower Tin Mountain Formation. Scale bar = 500 microns.

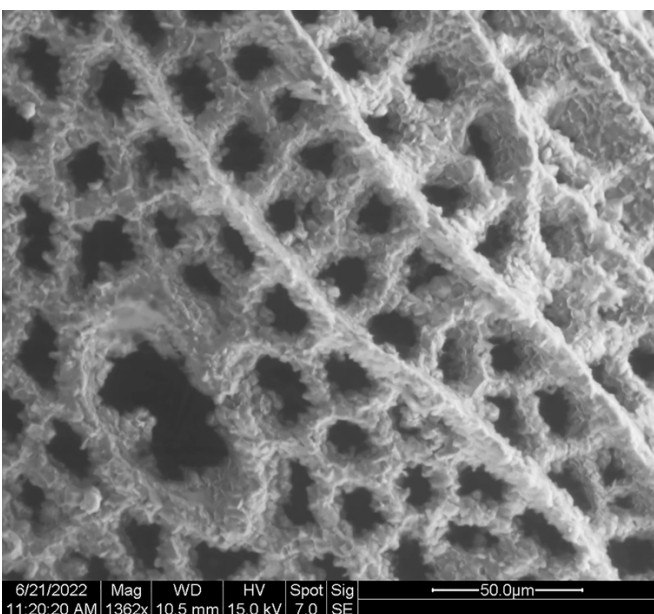

**Figure 3.** *Kirkbya panamintensis* nov. sp., detail of right valve. Holotype, IGM 5011(1). Scanning electron micrograph (secondary electrons). Crystal encrustations on the valve network are quartz crystals. Scale bar = 50 microns.

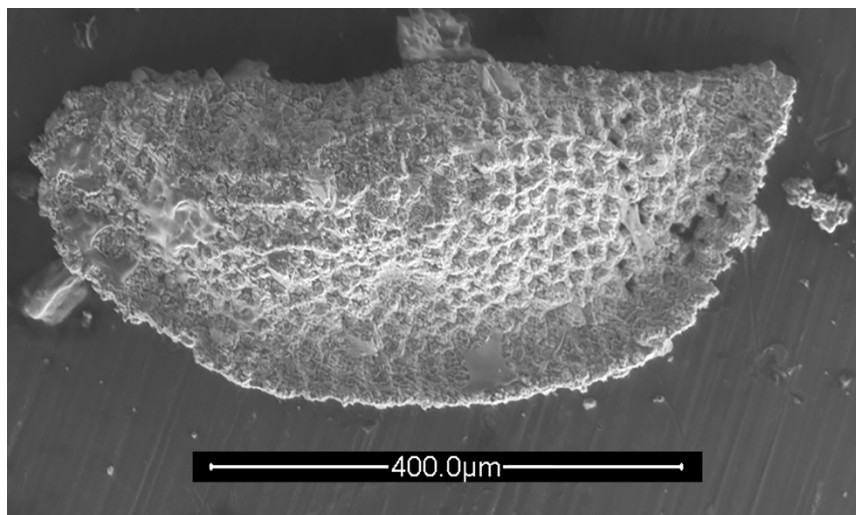

**Figure 4.** *Kirkbya panamintensis* nov. sp., left valve, lateral view. IGM 5011(2). Scanning electron micrograph (secondary electrons). Field sample MM-81-15; 29 m above the base of the Tin Mountain Limestone. Scale bar = 400 microns.

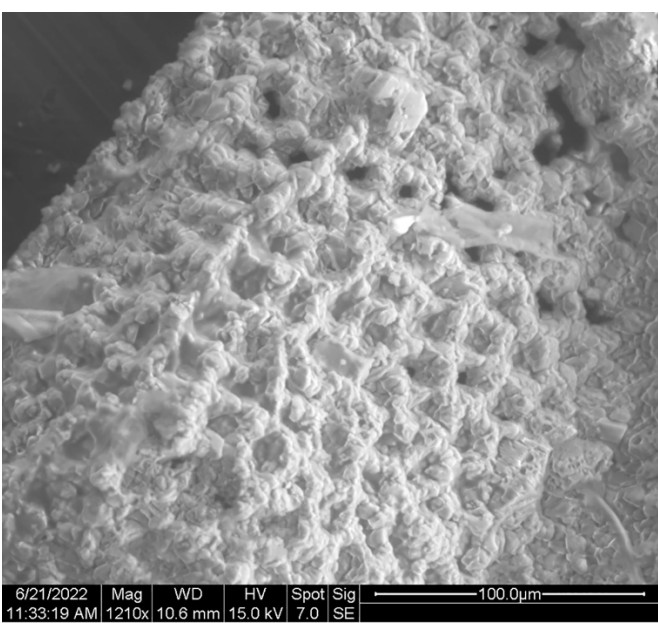

**Figure 5.** *Kirkbya panamintensis* nov. sp., detail of left valve. IGM 5011(2). Scanning electron micrograph (secondary electron). Image shows the occlusion of reticulate valve surface pores by silica (quartz) mineralization. Scale bar = 100 microns.

**Holotype:** IGM 5011(1).

**ZooBank entry:** urn:lsid:zoobank.org:act:9035EC9C-CA78-490D-8703-41552FC13D9A.

**Etymology:** The species name refers to the Panamint Mountains, California.

**Diagnosis:** A kirkbyan ostracod with elongate valves (height–width ratio 0.44), ornamented with fine riblets that occasionally come into contact. A reticulate, porous network constitutes most of the C surfaces. The prominent kirkbyan pit has a rim. The VB frill develops subdued radial ridges (marginal pore canals) that extend from the reticulate network.

**Description:** The species reaches approximately 0.9 mm in length and 0.4 mm in height.

**Discussion:** *Kirkbya panamintensis* nov. sp. is distinguished from *Kirkbya parva* Bushmina, 1975 [12] by having a more elongate valve (Figure 2). *K. parva* and *Kirkbya* sp. of Green [25] from the Banff Formation (? = *K. parva*) have height–width ratios ranging from

0.51–0.55, in contrast to the *Kirkbya panamintensis* nov. sp. ratio of 0.44. The kirkbyan pit, also known as the adductorial pit (Figure 3), indicates the position of the adductor muscle attachment. *Kirkbya panamintensis* nov. sp. is similar to the Late Permian Iranian ostracod *Kirkbya brandneri* (Kozur and Mette, 2006 [26]), but lacks the single prominent oblique ridge above the kirkbyan pit of the latter species. A single central horizontal ridge is seen in the Middle Devonian kirkbyid ostracod *Amphizona*; the feature is missing in very young *Amphizona* instars. The Carboniferous was an important time of ostracod diversification, especially for the kirkbyacean ostracods. Kirkbyacean ostracods alone are represented by more than a dozen genera during this time, and *Kirkbya panamintensis* nov. sp. adds to this record of diversification.

**Age and Locality Information:** Tin Mountain Limestone, (Lower Carboniferous/Lower Mississippian/Kinderhookian-early Osagean); 350–358.9 Ma.

**Stratigraphic position:** Two specimens are reported here. Field sample MM-81-15; 29 m above the base of the Tin Mountain Limestone. A second specimen (holotype) was recovered from the Lundin estate Tin Mountain Formation acid maceration sample. The Lundin sample specimen is from the lower Tin Mountain Formation, but its exact stratigraphic position is unknown.

**Collection data:** The ostracod from field sample MM-81-15 (5011(2)) was recovered by hydrochloric acid dissolution in June 2022 from rocks collected in fall 1981 from the north side of Lost Burro Gap, Inyo County, California, under a collecting permit #A9015 issued on December 31, 1981, by Ranger Richard S. Rayner, United States Department of the Interior, National Park Service, Death Valley National Monument, Death Valley, California 92328, USA. Specimens reside in the Institute of Geology Museum, Departmento de Paleontología, Cuidad Universitaria, México.

| Order Podocopida Sars, 1866 [27] |
| :---: |
| Suborder Podocopina Sars, 1866 [27] |
| Superfamily Bairdioidea Sars, 1887 [28] |
| Family Bairdiidae Sars, 1887 [28] |
| Genus *Bairdia* McCoy, 1844 [29] |
| *Bairdia quasilecta* Bushmina, 1975 [12]: Figure 6 |

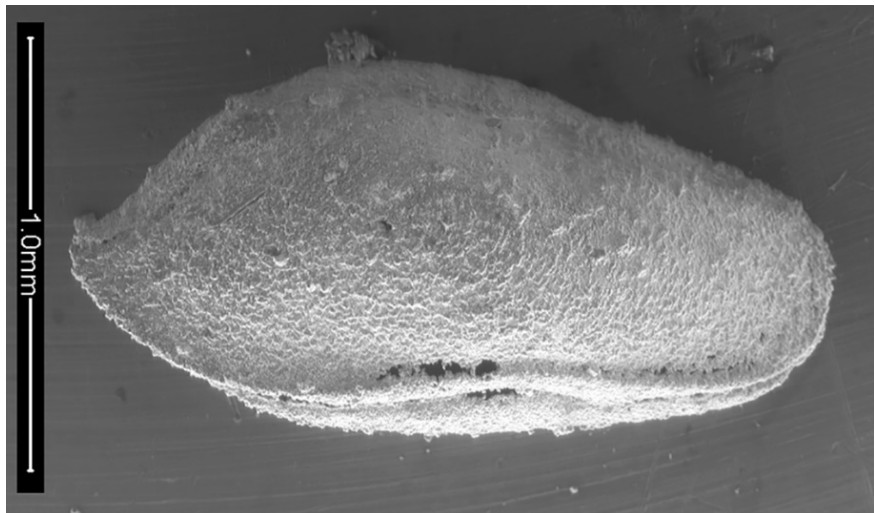

**Figure 6.** *Bairdia quasilecta* Bushmina, 1975, right valve, oblique lateral view. IGM 5011(3). Scanning electron micrograph (secondary electrons). Specimen is slightly rotated on the long axis to show the ventral border. Scale bar = 1mm.

**Material:** One specimen, both valves united and intact. IGM 5011(3). Field sample ME-8; 41.2 m above the base of the Tin Mountain Limestone.

**Description:** An elongate species of *Bairdia* with strong DB overlap of the LV over the right valve, and an open VB between the two valves that, with rounded VB selvages on both valves giving the impression of a parted pair of lips.

**Discussion:** *Bairdia quasilecta* Bushmina, 1975 [12] was originally described from the right bank of Kamenka Creek, Kolyma River region, Russian Far East. The Russian ostracod is a nearly exact match to the Tin Mountain Limestone specimen illustrated here. The California specimen somewhat resembles *Bairdia* sp. aff. *B. egorovi* Sohn, 1960 [30] as illustrated by Green [25], but the latter has a blunter, less pointed PB. The Tin Mountain specimen has some similarity to *Bekena pecki* (Morey, 1935 [31]) as illustrated in Sohn [30], but the valve is more elongate and the anterior margin is more acuminate.

**Age and Locality Information**: As for previous.

**Collection data:** Recovered by acid maceration of field sample ME-8.

---

*Bairdia* sp. cf. *B. orientalis* Bushmina, 1975 [12]: Figure 7

---

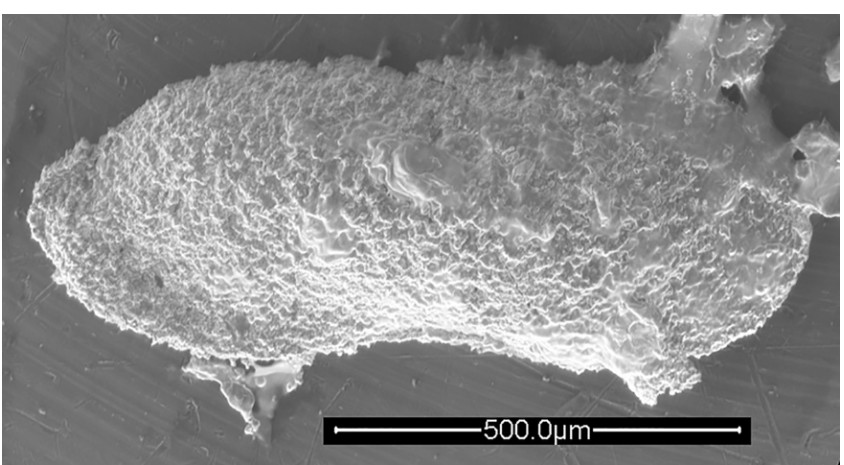

**Figure 7.** *Bairdia* sp. cf. *B. orientalis* Bushmina, 1975, right valve, lateral view. IGM 5011(4). Scanning electron micrograph (secondary electrons). Scale bar = 50 microns.

**Material:** One specimen. IGM 5011(4). Field sample ME-8; 41.2 m above the base of the Tin Mountain Limestone.

**Description:** An elongate species of *Bairdia* with a curved, flattened PVM. The hinge is long and straight, whereas the VB is strongly convex. The AVB is rounded.

**Discussion:** This California species resembles *Bairdia orientalis* Bushmina [12].

**Collection data:** As for previous.

**Age and Locality Information**: As for previous.

---

Genus *Rectobairdia* Sohn, 1960 [30]

---

*Rectobairdia* sp. cf. *R. legumen* (Jones and Kirkby, 1886 [32]): Figure 8

---

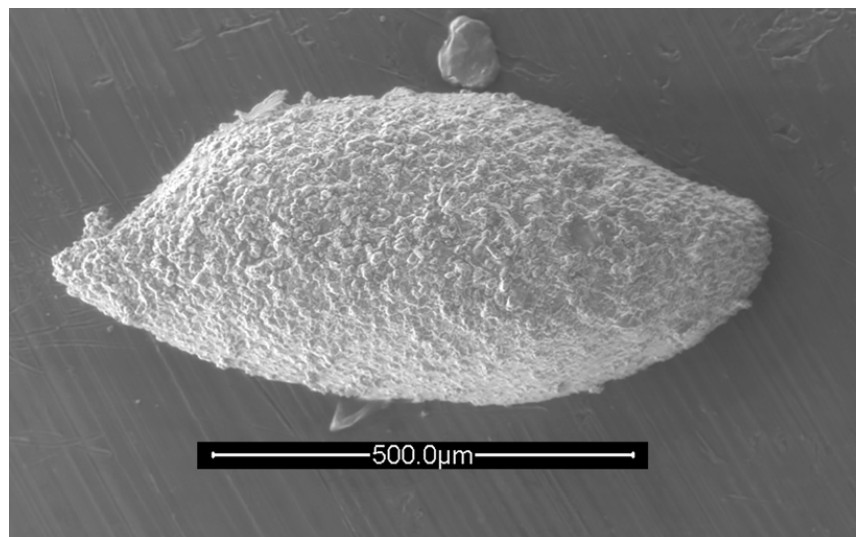

**Figure 8.** *Rectobairdia* sp. cf. *R. legumen* (Jones and Kirkby, 1886), left valve, lateral view. IGM 5011(5). Scanning electron micrograph (secondary electrons). Scale bar = 500 microns.

**Material:** Three specimens. Field sample ME-8; 41.2 m above the base of the Tin Mountain Limestone.

**Description:** The upper halves of the valves have a trapezoidal aspect. The DB is very slightly curved to straight. The ADB is straight, then bends to constitute a pointed anterior corner; the anterior corner is below the valve midline. The PDB is gently curved, and the PVB is more broadly arched; the posterior corner is above the midline. Maximum valve height is two thirds the distance from the anterior corner tip.

**Discussion:** This species of *Rectobairdia* bears similarity to *Rectobairdia legumen* (Jones and Kirby, 1886 [32]) as illustrated (as *Bairdia legumen*) by Bushmina [12], except that in the Tin Mountain specimen the posterior end of the valve turns up slightly, in a character state intermediate between Bushmina's [12] *B. legumen* and *B. gibbera*.

**Collection data:** As for previous.

**Age and Locality Information:** As for previous.

| Genus *Ceratobairdia* Sohn, 1954 [33] |
| :---: |
| *Ceratobairdia* sp.: Figures 9 and 10 |

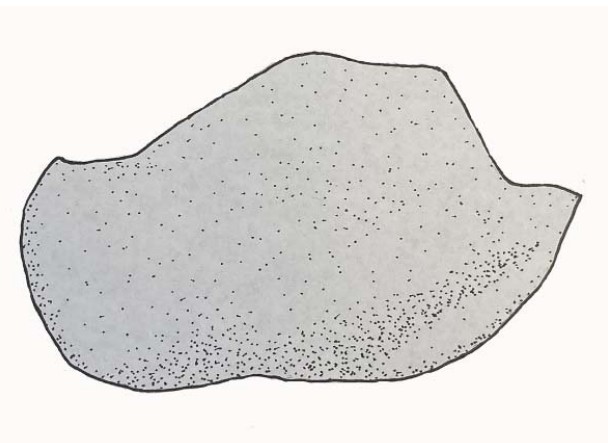

**Figure 9.** *Ceratobairdia* sp., right valve, lateral view. IGM 5011(6). Width of valve 1.2 mm. Image credit: Mark McMenamin.

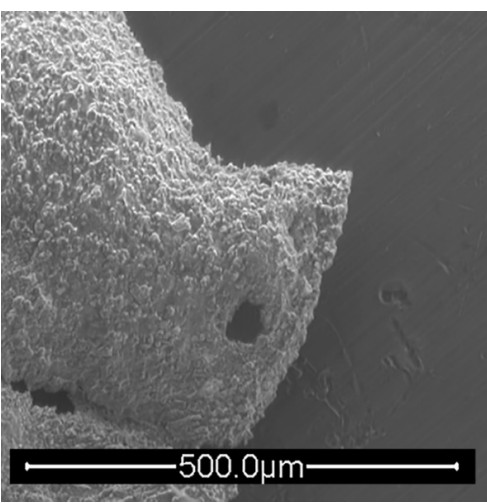

**Figure 10.** *Ceratobairdia* sp., right valve, lateral view. IGM 5011(6). Scanning electron micrograph (secondary electrons). Detail of anterior corner spine. Scale bar = 500 microns.

**Material:** One specimen. IGM 5011(6). Field sample ME-8; 41.2 m above the base of the Tin Mountain Limestone.

**Description:** This ceratobairdiid has a relatively short, straight hinge (DB), a steep, straight ADB, a more gently inclined PDB, and a PVB that flares out broadly. The VB is slightly concave. The anterior corner spine (Figure 10) is prominent and broad. The posterior corner spine is very broad, and curves gently to a pointed tip.

**Discussion:** This species has the most distinctive profile of all the ostracods reported here from the Tin Mountain fauna, with its trapezoidal DB and inclined dorsal profile. The Tin Mountain specimen bears some resemblance to *Ceratobairdia wordensis* (Hamilton, 1942 [34]) as illustrated by Sohn [30] but has a slightly concave rather than convex hinge line.

**Collection data**: As for previous.

**Age and Locality Information**: As for previous.

| Family Bairdiocyprididae Shaver, 1961 [35] |
| :---: |
| Genus *Silenites* Coryell and Booth, 1933 [36] |
| *Silenites* sp.: Figure 11 |

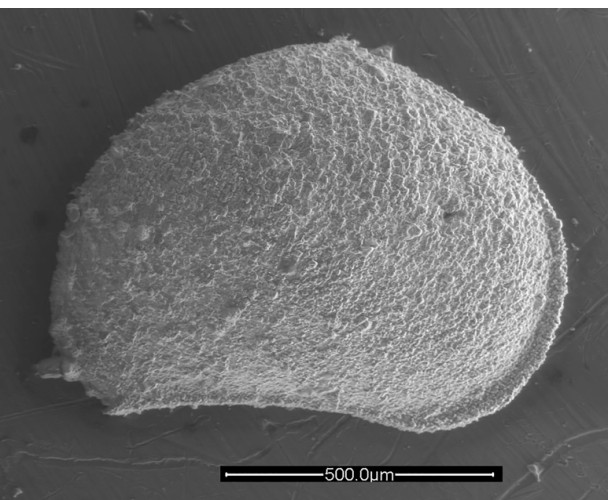

**Figure 11.** *Silenites* sp., left valve, lateral view. IGM 5011(7). Scanning electron micrograph (secondary electrons). Scale bar = 500 microns.

**Material:** Two specimens. Field sample MM-81-39; 39.3 m above the base of the Tin Mountain Limestone.

**Description:** A smoothly rounded silenitid ostracod that is almost circular in outline, except at the VB where the valves are smoothly concave. The species has a narrow VB flange, and a comparatively wider PVB flange. The DB flange narrows and disappears, and the AB flange appears to be absent.

**Discussion:** This California species of *Silenites* has a symmetrical DM with a peak at the valve midline. The DM slopes away from this peak in a straight line, until forming on both sides a slight inflection. The AB and PB are smoothly curved and fairly wide. This species is less elongate and has a less angular profile than seen in *Silenites* sp. cf. *S. warei* as illustrated by Green [25]. The Tin Mountain specimen is also less elongate than *Silenites* sp. of Tanaka et al. [37].

The nearly circular outline and narrow flanges of *Silenites* sp. are similar to those of the modern ostracod *Notodromas trulla*. The Notodromadinae have an inverted countershading color scheme because they swim (at the surface of the water) upside down for neustonic feeding.

**Collection data**: As for previous.

**Age and Locality Information**: As for previous.

---

Family Acratiidae Gründel, 1962 [38]

---

Genus *Acratia* Delo, 1930 [39]

---

**Discussion:** Green [25] noted that in the basal Banff Formation (Jasper), a species very similar to *Acratia similaris* Morey ranges from mid-unit 3 to upper unit 4, where it is replaced by *Acratia similaris* proper and *Acratia fabaeformis* by mid-unit 5, with *A. fabaeformis* ranging into the basal part of unit 6.

---

*Acratia* sp. A: Figure 12

---

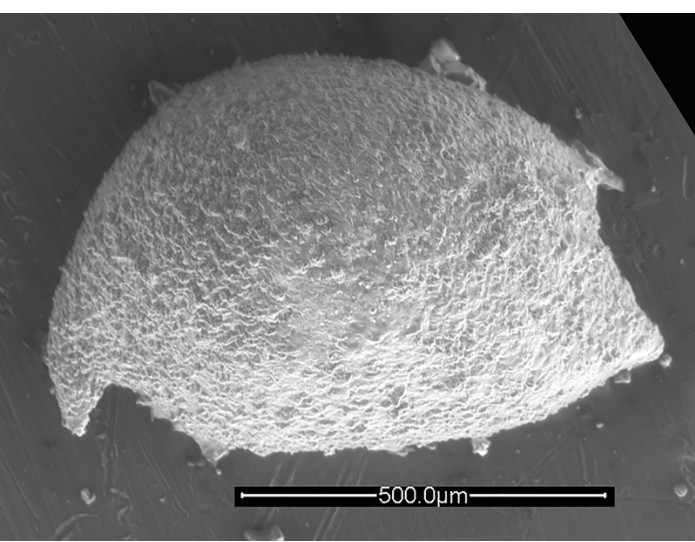

**Figure 12.** *Acratia* sp. A, left valve, lateral view. IGM 5011(8). Scanning electron micrograph (secondary electrons). Scale bar = 500 microns.

**Material:** One specimen. IGM 5011(8). Field sample ME-8; 41.2 m above the base of the Tin Mountain Limestone.

**Description:** This acratiid ostracod has in inflated valve in profile, with a highly arched DB and a convex VB. The anterior corner spine tapers to a downward pointing 'beak'-like structure. The PDB is broken but appears to smoothly curve to a nipple-shaped posterior corner spine. Both corner spines are well below the valve midline and close to the VB.

**Discussion:** This species of *Acratia* was first reported by McMenamin and Witterschein [11]. It is characterized by its pronounced 'beak' (anterior downward-pointing corner spine) and is slightly concave on its posterior margin. The DB is smoothly arched and highly vaulted.

**Collection data**: As for previous.

**Age and Locality Information**: As for previous.

*Acratia* sp. B: Figure 13

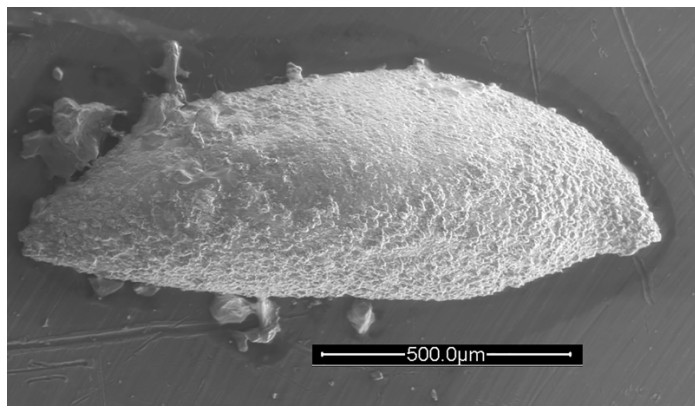

**Figure 13.** *Acratia* sp. B, right valve, lateral view. IGM 5011(9). Scanning electron micrograph (secondary electrons). Scale bar = 500 microns.

**Material:** One specimen. IGM 5011(9). Field sample ME-8; 41.2 m above the base of the Tin Mountain Limestone.

**Description:** An elongate acratiid ostracod with a gently convex DB, a fairly steeply inclined PDB, and a straight to slightly convex VB. The anterior corner spine (although broken in this specimen) points forward rather than pointing down. There appears to be a pair of small posterior corner spines, although this may be due to breakage of the specimen.

**Discussion:** This species of *Acratia* resembles *Acratia* sp. 1 of Tanaka et al. [37] but has a more smoothly curved profile and a less sharply pointed 'beak' (= anterior downward-pointing corner spine).

**Collection data**: As for previous.

**Age and Locality Information**: As for previous.

*Acratia* sp. C: Figure 14

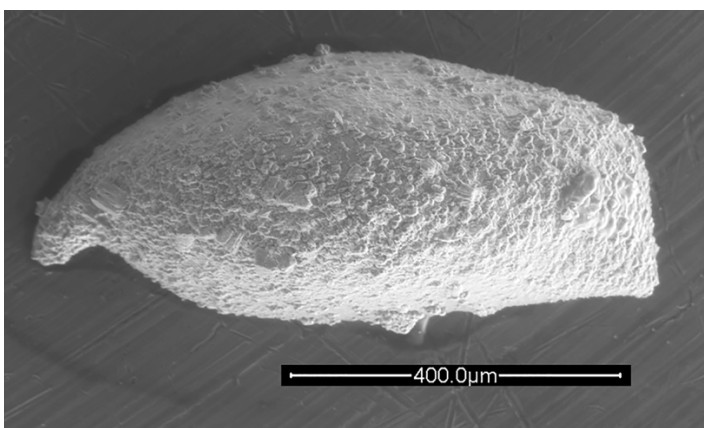

**Figure 14.** *Acratia* sp. C, left valve, lateral view. IGM 5011(10). Scanning electron micrograph (secondary electrons). Scale bar = 400 microns.

**Material:** One specimen. IGM 5011(10). Field sample ME-8; 41.2 m above the base of the Tin Mountain Limestone.

**Description:** This species of *Acratia* has the pointed 'beak' (anterior down-and-forward pointing anterior corner spine) characteristic for the genus, plus a very blunt, nearly vertical PB, that terminates in the PVB in a broad posterior corner spine that has sides that are almost at right angles to one another.

**Discussion:** The straight, nearly vertical PB of this species is both unique and unusual for acratiids.

**Collection data:** As for previous.

**Age and Locality Information:** As for previous.

---

*Acratia* sp. D: Figures 15 and 16

---

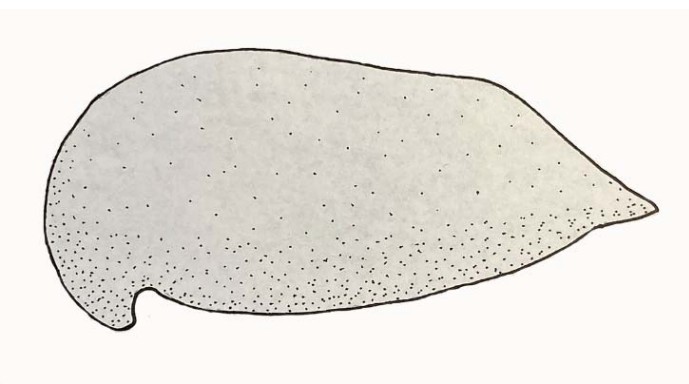

**Figure 15.** *Acratia* sp. D, left valve, lateral view. IGM 5011(11). Width of valve ~1 mm. Image credit: Mark McMenamin.

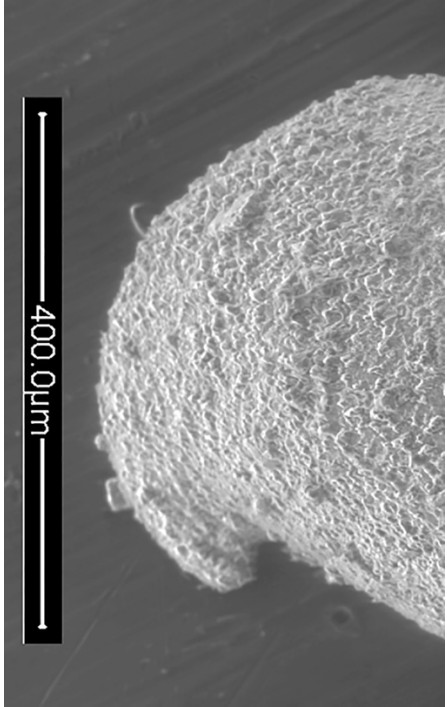

**Figure 16.** *Acratia* sp. D, left valve, close up of anterior of valve. IGM 5011(11). Scanning electron micrograph (secondary electrons). Scale bar = 400 microns.

> **Material:** Three specimens. Field sample ME-8; 41.2 m above the base of the Tin Mountain Limestone.
>
> **Description:** An acratiid ostracod with an AB arched similar to an arc of a circle (Figure 16), an anterior spine that points backwards, and a PDB and PVB that are drawn out into pointed posterior corner spine that points backwards and is positioned approximately along the valve midline. The DB is straight, and the ventral border is broadly convex.
>
> **Discussion:** This elongate species of *Acratia* has a bulging AB and a nearly straight DB (hinge area). The species has a more curved 'beak' (anterior downward-pointing corner spine), than *Acratia sinuata* (Kozur, 1991 [40]), as described by Tarnac et al. [41]. The California species has a distinctive, acuminate valve posterior somewhat comparable to that of *Acratia* sp. 1 of Tanaka et al. [37].
>
> **Collection data:** As for previous.
>
> **Age and Locality Information**: As for previous.

**Funding:** This research received no external funding.

**Acknowledgments:** The author wishes to thank S. M. Awramik, D. Pierce, K. Davis, D. L. Schulte McMenamin, Claire Pless, L.E. Witterschein and two anonymous reviewers for assistance with this project. Specimens collected via permit #A9015 issued on December 31, 1981, by Ranger Richard S. Rayner, United States Department of the Interior, National Park Service, Death Valley National Monument, Death Valley, California 92328. A copy of the permit is on file. Specimens reside in the Institute of Geology Museum, Departmento de Paleontología, Cuidad Universitaria, México.

**Conflicts of Interest:** The author declares no conflict of interest.

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
