# Peer review of "Early Carboniferous Ostracods (Crustacea) from Death Valley, California, USA"

_geosciences, doi:10.3390/geosciences12080300_

Round 1
Reviewer 1 Report
Abstract: The abstract is written in a verbose style. As a result, the message is not as clear as it could be. The abstract does not offer the topic of the paper in a broader context.
· Introduction. the introduction needs to be modified and elaborated. The authors need to make some more general statements here as to why their study is of interest to readership and what we can learn in general with regard to ostracod assemblages in shallow marine carbonate ramp habitats along the northern shores of Pangea.
· The entire introduction chapter is not set up as a typical introduction chapter. It’s a Systematic Paleontology chapter!
· The objectives of the paper have to be stated clearly in the introduction.
· The text spends much space explaining the systematic paleontology of ostracod assemblages from the Lower Carboniferous (Lower Tin Mountain Limestone), but readers will know all this. You need to focus on the relationship of this fauna with shallow marine carbonate ramp habitats along the northern shores of Pangea at relatively low latitudes. You mentioned this issue in several places in the article, but you did not give any explanation. This topic can be included as a separate chapter as a discussion in the article.
· Line 56 to 66 can be emplaced in discussion chapter
· 107-109 “The formation represents a shallow marine limestone that was deposited in mudflats, lagoons, and offshore carbonate sand bars”. These results need more explanations and more reasons.
I hope the author find the comments/suggestions and I would encourage them to address carefully the comments as they should augment substantially the quality and potential impact of this study.
Author Response
Replies to Reviewer 1
- The style of the abstract has been improved.
- The Introduction has been split into two sections, with enhanced discussion regarding ostracod assemblages in shallow marine carbonate ramp habitats. The objectives of the paper are now clearly stated in the Introduction.
- The Introduction is not a Systematic Paleontology section—that comes later in the paper.
- Lines 56-66 have been moved to a new section.
- Regarding original lines 107-109, more context (with additional references) has been provided for the depositional environment and paleoecology of the ostracods.
Reviewer 2 Report
The manuscript is well written and organized. The introduction provides sufficient background, materials and methods section is well explained in detail, the results are clearly presented. Species description and illustration is appropriate and complete. All figures are of high quality and figure captions are adequate. Consequently, this manuscript is suitable for publication and is mature enough to be published as it stands.
Author Response
Thanks to Reviewer #2 for their comments.